# Biosynthesis of Copper Nanoparticles with Medicinal Plants Extracts: From Extraction Methods to Applications

**DOI:** 10.3390/mi14101882

**Published:** 2023-09-30

**Authors:** Aurora Antonio-Pérez, Luis Fernando Durán-Armenta, María Guadalupe Pérez-Loredo, Ana Laura Torres-Huerta

**Affiliations:** 1Departamento de Bioingeniería, Escuela de Ingeniería y Ciencias, Tecnologico de Monterrey, Campus Estado de México, Av. Lago de Guadalupe KM 3.5, Margarita Maza de Juárez, Atizapán de Zaragoza, Ciudad López Mateos 52926, Mexico; a.antonio@tec.mx (A.A.-P.); guadalupe.perez@utfv.edu.mx (M.G.P.-L.); 2VIB-VUB Center for Structural Biology, Vlaams Instituut voor Biotechnologie, Pleinlaan 2, 1050 Brussels, Belgium; luis.fernando.duran.armenta@vub.be; 3Structural Biology Brussels, Vrije Universiteit Brussel, 1050 Brussels, Belgium; 4División Académica de Tecnología Ambiental, Universidad Tecnológica Fidel Velázquez, Av. Emiliano Zapata S/N, El Tráfico, Nicolás Romero C.P.54400, Mexico

**Keywords:** nanoparticles, green synthesis, flavonoids, polyphenols, terpenoids, extracts, traditional medicine

## Abstract

Copper nanoparticles (CuNPs) can be synthesized by green methods using plant extracts. These methods are more environmentally friendly and offer improved properties of the synthesized NPs in terms of biocompatibility and functional capabilities. Traditional medicine has a rich history of utilization of herbs for millennia, offering a viable alternative or complementary option to conventional pharmacological medications. Plants of traditional herbal use or those with medicinal properties are candidates to be used to obtain NPs due to their high and complex content of biocompounds with different redox capacities that provide a dynamic reaction environment for NP synthesis. Other synthesis conditions, such as salt precursor concentration, temperature, time synthesis, and pH, have a significant effect on the characteristics of the NPs. This paper will review the properties of some compounds from medicinal plants, plant extract obtention methods alternatives, characteristics of plant extracts, and how they relate to the NP synthesis process. Additionally, the document includes diverse applications associated with CuNPs, starting from antibacterial properties to potential applications in metabolic disease treatment, vegetable tissue culture, therapy, and cardioprotective effect, among others.

## 1. Introduction

A nanoparticle (NP) is a small particle with a diameter ranging from 1 to 100 nm. NPs are the most widely produced nanomaterial due to their extensive industrial applications, which include aerospace, automotive, agriculture, cosmetics, textiles, food packaging, and additives, among many others [1,2,3,4]. In recent years, the rapid growth of the global market for nanomaterials has incentivized research on the potential use of NPs for diagnostics, biomarkers, drug development and delivery, cell labeling, cancer therapy, and antiviral and antimicrobial agents [5,6]. The latter have gained more interest due to the growing threat of antibiotic resistance. NPs have been integrated as antibacterial agents in the development of different products, such as creams to treat burns and injuries, and skincare products [7,8]. Remarkably, two COVID-19 vaccines containing lipid NPs were approved by both the US Food and Drug Administration (FDA) and the European Medicines Agency (EMA). Lipid-based NPs carry the mRNA encoding for the spike protein of SARS-CoV-2 and play a key role in protecting and delivering the cargo [9]. In the field of diagnostics, various methods utilizing NPs have been developed. These methods involve the detection of specific agents or the diagnosis of diseases by functionalization of surfaces [10]. Likewise, metal NPs have been utilized in bone treatments, such as promoting osteoblast formation to address bone inflammation and bone regeneration [11].

Classical NP synthesis process involves the use of hazardous materials, including toxic, corrosive, and explosive substances, and utilizes both chemical and physical methodologies. Furthermore, the proper disposal of potentially hazardous by-products and waste becomes imperative to mitigate the already significant environmental impact of the process. These risks primarily depend on the selected precursor, for instance, some copper salts are more dangerous than others. Moreover, classical methods require specialized equipment with high energy costs, which presents significant challenges in establishing and maintaining optimal operation conditions [1,12].

The need for more cost-effective and sustainable alternatives for NP synthesis has drawn interest in biological methods. Among these, green synthesis has emerged as a straightforward, rapid, and sustainable approach. In the synthesis process with plants, alcoholic or aqueous extracts are used (Figure 1), combined with methods such as microwaves or magnetic agitation [13].

Among the most frequently utilized metals for synthesizing NPs are silver (Ag), gold (Au), palladium (Pd), iron (Fe), platinum (Pt), nickel (Ni), cobalt (Co), and copper (Cu) and their oxidized forms, such as iron oxide (Fe_3_O_4_), zinc oxide (ZnO), copper oxide (CuO), among others [4,14]. From this list, copper has become a metal of interest due to its natural presence in biological systems, antimicrobial properties, low toxicity, and higher cost-effectiveness when compared to Au and Ag. In this review, we present an overview of the green synthesis methods applied to CuNPs, particularly those using extracts from medicinal plants.

## 2. Copper NPs

Copper is an essential trace mineral naturally present in some foods. In various metabolic pathways, it functions as a cofactor for numerous enzymes involved in energy production, and biosynthesis of connective tissue, neurotransmitters, collagen, and red blood cells [15,16]. Copper also plays a significant role in supporting normal brain development, immune functions, dismutation of superoxide radicals, skin regeneration, angiogenesis, and healing processes [16,17,18]. However, both copper deficiency and excess can cause illnesses in humans. Moreover, the antiseptic properties of copper have been documented extensively throughout history, with records as early as 3000 BC [19]. Copper-based NPs have been proven to be effective against a broad range of clinically relevant pathogenic microorganisms [20,21].

Despite the widespread use of NPs, there are still concerns regarding their potential risks to both the environment and human health [1,6]. Although the toxicity of copper is considered low for humans, there is limited research available on the toxicity of CuNPs [15]. Recent studies have shown copper nanoparticle-induced toxicity in the kidney, liver, and spleen [22], and the reproductive system in mice [23]. Moreover, CuNPs induced apoptosis and suppressed cellular proliferation in human extravillous trophoblast cells [23] and SW480 cells [24]. It is important to note that the characteristics and potential toxicity of CuNPs are influenced by several factors, including the composition of the biological agent, the precursor salt used, the synthesis conditions, and the resulting particle size [1]. Although a complex and challenging task, it is essential to optimize the synthesis process to obtain NPs with high antimicrobial activity and low toxicity. Characterization of CuNPs will also shed light on their physical properties as well as the potential mechanism(s) for toxicity *in vivo*.

## 3. Green Synthesis of Nanoparticles

Conventional methods for NP synthesis can be classified into two categories: top-down and bottom-up. The key distinction between these categories lies in their approach, the main difference being the starting material to synthesize NPs. In top-down methods, the bulk material is broken down into NPs using different physical, chemical, and mechanical processes like laser ablation, mechanical milling, and sputtering. Bottom-up approaches like vapor deposition, chemical reduction, spray pyrolysis, and biological/green synthesis involve the utilization of atoms or molecules that act as the starting material in the formation of NPs [12,25]. Both top-down and bottom-up approaches rely on different physical and chemical methods to prepare and stabilize metallic NPs. However, many of the current methods require expensive specialized equipment and very high energy input [12]. The environmental impact of conventional NP synthesis is further aggravated due to the use of harmful chemicals as reagents and the production of dangerous by-products [6,12,25].

Among the bottom-up approaches, green synthesis of NPs has become a growing trend, as it offers solutions to the major disadvantages of conventional methods, mainly the high costs, toxicity, safety issues, and environmental impact. The integration of nanotechnology and green chemistry has resulted in novel biosynthetic routes to produce NPs in clean, non-toxic, and eco-friendly processes that utilize natural sources like plants and microorganisms [12,25,26].

The components for nanoparticle biosynthesis or green synthesis include extracts from the different parts of a plant, such as the roots, stems, leaves, fruits, peels and seeds, which have varying benefits in this process. Metabolites from microorganisms (bacteria, actinomycetes, yeast, algae), viral particles, and even biomass waste are utilized [1]. The principle of green synthesis relies on the bioactive reducing properties of plant metabolites like terpenoids, alkaloids, and proteins. In addition, poly-hydroxyl groups in secondary metabolites are instrumental in the synthesis of NPs through their ability to reduce metal ions [6,27].

The synthesis of NPs occurs as a natural process in microorganisms and plants. There are two main ways in which biogenic synthesis can be performed by living organisms: intracellular (endogenous) and extracellular (exogenous) synthesis. Endogenous biosynthesis relies on the ability of microorganisms and plant cells to hyper-accumulate metals from their medium. Metals are reduced within the cell cytosol and retained as nanometric particles. For example, there is research indicating the ability of plants to hyper-accumulate and eliminate heavy metals, which are extremely toxic even at low concentrations [1,25,28]. Exogenous biosynthesis relies on the secretion of secondary metabolites by plant roots under metal stress. These metabolites chelate metallic ions and reduce them into nanoscale particles [1]. While the precise mechanisms underlying biogenic synthesis are not yet fully understood, ongoing research efforts are dedicated to enhancing our comprehension of the processes controlling nanoparticle formation. These studies have yielded progress in the development of novel procedures to synthesize and refine control over NP morphology and size, as well as in identifying new and promising applications. This depends on the interaction of the capping agent (biomolecules or secondary metabolites) with a particular facet of a metal or metal oxide crystal. In nature, there are abundant biological resources (both plants and microorganisms, among others) that can potentially be used as capping agents for the synthesis of NPs.

Depending on the natural source used for the synthesis, the method is adjusted to make it more efficient and take advantage of favoring the formation of NPs. For example, if using microorganisms, they can be readily cultivated in liquid media and subsequently use microbial culture filtrates (extracellular and intracellular) as a reducing agent for nanoparticle production [29]. Regarding green synthesis employing plant extracts, the synthesis workflow can be summarized into six key steps: plant part selection, washing, drying, mechanical grinding, high-temperature solvent mixing, and, ultimately, filtration of the solvent extract [1,30].

The synthesis of metal nanoparticles (NPs) involves three primary components: reducing and stabilizing agents, along with a solvent medium. In the context of green synthesis, many phytocomponents act as reductors and stabilizers when mixed with the precursor metal salt in aqueous solution [26]. In the next section, we will review in detail how the extraction of plant components is carried out to obtain NPs. Briefly, solutions of metallic copper precursor salts are mixed with plant extracts under specific reaction conditions. The properties of the NPs are influenced by the conditions under which the synthesis takes place, including factors like the selected precursor salt, pH, temperature, precursor-to-extract ratio, and more. The stages proposed thus far for the green synthesis of NPs are as follows: First, the ionic form of copper is generated. Ionic copper is susceptible to receiving electrons from plant compounds with reducing capacity. Consequently, a reduction process converts the ionic form into a neutral atom, which leads to aggregation and growth, followed by subsequent oxidation and coating. The coating is achieved through interactions with plant compounds that retain their reactivity. These interactions contribute significantly to stabilizing the resulting NPs. A representation of the general process of green synthesis of CuNPs is shown in Figure 2, in which the main contributions of the complex mixture of phytocomponents from plant extracts are highlighted. This synthesis process is described in greater depth in Section 7.

A diverse range of medicinal plants have been recorded for their application in the production of CuNPs. The remarkable diversity of components found in medicinal plants contributes to the achievement of a wide spectrum of shapes, sizes, and properties in the resulting NPs. Regarding shape, NPs have been described as spherical, nanocomposite, agglomerated, rods, hexagonal, triangular, cylindrical, prismatic, and amorphous, among others.

## 4. Potential of Medicinal Plants

Among the alternatives reported so far, nanoparticle synthesis using plant extract offers advantages over the use of fungi, algae, and bacteria [25]. Plants are the reservoirs of naturally occurring chemical compounds and structurally diverse bioactive molecules (Figure 1) [31,32].

Traditional herbal medicine (or alternative herbal medicine) is the use of plant-derived substances obtained with little or no industrial processing to treat diseases within regional healing practices. Medicinal plants were discovered and empirically formulated through practical observation by ancient civilizations, and this has continued for millennia. Today, they are commonly administered instead of or together with conventional pharmacological drugs. Plants that are known to be medicinal can be found everywhere and our lives are entwined with a diverse range of plants, in addition to the fact that many of these plants are included in food. Traditional herbal medicine has been used to treat or prevent many diseases and ailments [33]. The most ancient recorded proof of using medicinal plants to create remedies comes from Sumer, while the following evidence corresponds to traditional Chinese medicine, with prescription records dating back to 2500 BC [34]. Subsequently, as they developed, various cultures also made important contributions to the development of traditional medicine. For example, we can mention some examples such as Japanese *kampo*, Tibetan, Indian, and Mexican medicine. Below, we describe some characteristics of traditional medicine from some cultures, mainly to highlight its diversity and as an important source of knowledge on plants.

For millennia, traditional Chinese medicine (TCM) has held the distinction of being one of the earliest therapeutic remedies, having played a substantial part in the management of numerous diseases throughout history. This traditional medicine includes herbal medicine, acupuncture, moxibustion, and massages. Chinese herbal medicine has been used as both a pharmaceutical and dietary supplement, mainly in antibacterial and anticancer treatments [35]. More than 11,000 species of medicinal plants have been recorded in various TCM-related pharmacopeias [36]. Some representative Chinese herbs that are consumed as vital food supplements or as folk medicines include *Cinnamomum cassia*, *Ephedra sinica*, *Zingiber officinale*, *Angelica sinensis*, *Astragalus membranaceus*, *Ganoderma lucidum*, *Panax notoginseng*, and *Zingiber officinale* [37,38]. Variation and adaptation of traditional Chinese occurred in nearby countries, such as Korea, Japan, and Vietnam.

Tibetan medicine (TM) ranks as the second-largest traditional medicine system in China. The origins of TM date back 2500 years. The principle of this traditional medicine relied mainly on the use of natural herbal remedies, particularly for treating wounds, and a combination with ritual practices [39]. Treatment consists of medicines usually prepared from plants, rare minerals or animals, physical therapies such as massages or baths, regulation of lifestyle and diet, and spiritual practices [40]. To date, the Tibetan medicine system has had around 3105 distinct types of natural medicines [41] catalogued, and has gained extensive application in managing chronic illnesses, including conditions like rheumatism, high-altitude polycythemia, cholecystitis, hepatitis, and gastritis [42]. The National Medical Products Administration has approved and registered over 200 preparations of Tibetan medicine. A great abundance of chemical constituents is found in Tibetan medicine preparations. *Qishiwei zhenzhu* pills (the most well-known preparation of TM) contain around 70 medicinal materials, which results in a multitude of targets and pathways of action. This complexity contributes to the intricacy of the mechanisms of action of Tibetan medicine. Some examples of plants mainly used in TM are *Rhodiola crenulata*, *Hippophae rhamnoides* L., *Terminalia chebula Retz*, and *Pterocephalus hookeri* [42].

Traditional Japanese medicine, which includes *kampo-yaku*, acupuncture, and acupressure, has been practiced for over 1500 years. Kampo medicine, the traditional herbal medicine of Japan, originated and underwent adaptations from Chinese herbal medicine [43]. Kampo medicine has great relevance in Japan and it is manufactured with good manufacturing practice (GMP) stipulated by Japanese law as well as for rigorous scientific investigation. The Japanese Ministry of Health, Labor, and Welfare has approved 294 kampo formulas for clinical use [44]. This traditional medicine utilizes pharmaceutical-grade multi-herbal treatments to attain a synergistic effect. A notable characteristic of many kampo medicines is that they are derived from a hot water extract composed of a blend of medicinal herbs. Due to the intricate nature of these medicinal herb combinations, it is more convenient to assign them specific names. Such is the case of *hochuekkito*, which is composed of around nine plants [45]. Typical kampo medicines are composed of five to nine different plants, with the most commonly used ones being *Glycyrrhiza uralensis radix* (Chinese licorice root), *Zingiber officinale rhizome* (ginger rhizome), *Poria cocos* (pachyme mushroom), and *Paeonia lactiflora* radix (Chinese peony root) [46]. Kampo medicines have been conventionally prescribed for numerous health conditions due to their promising effects against viral infections, acute and chronic pain, chronic hepatitis, allergic rhinitis, anemia, gastric cancer, hypertension, fever, hypercholesterolemia, and oral diseases/disorders, among others [46,47,48].

The traditional Indian system of medicine has its own distinct recognition as a traditional medicine: Ayurveda, yoga, Unani, Siddha, and homoeopathy (AYUSH). The system’s foundation is holistic, and its pharmacological methods are based on natural sources, such as plants, animals, or minerals [49]. In India, the culinary application of numerous medicinal species is widespread, and they are a part of the kitchen in every house. A few examples are giloe, ginger, cinnamon, black pepper, black cumin, turmeric, and garlic. Some of those herbs have proven immunomodulatory, antioxidant, and anti-infective properties [49]. On the other hand, the cultural and artistic components of the Kerala tradition are portrayed through the 10 sacred flowers of “Dasapushpam,” known in India for centuries [50].

Mexican culture has possessed extensive knowledge of traditional medicine that integrates the therapeutic methods of pre-Columbian civilizations, such as the Maya and Aztecs, that existed before the Spanish acme, in addition to more recent African and Spanish-Catholic influences. Today, Mexico has an extensive variety of medicinal plants, with over 23,400 vegetable species and 5000 different plant species being employed for medicinal purposes [51]. Digital libraries are an important resource of traditional medicine information in Mexico, encompassing approximately 1045 monographs dedicated to medicinal plants [52]. Some plants in traditional Mexican medicine widely utilized by at least four ethnic groups are *Aristolochia odoratissima*, *Citrus aurantium* L., *Equisetum myriochaetum*, *Justicia spicigera*, *Matricaria chamomilla*, *Piper auritum Kunth*, and *Sambucus nigra* [53]. Most common uses of traditional Mexican medicine are focused on conditions of the digestive system, culture-bound syndromes, genitourinary system, infectious and parasitic diseases, pregnancy, childbirth, and the respiratory system, among others [53,54,55].

Modern medicine laid its foundations on the traditional knowledge of the use of medicinal herbs. Important advances have been made in the identification of active compounds derived from plants that have been used in the development of specialized medicines. There is still much to explore in traditional medicine. However, to achieve acceptable, safe, and consistent efficacy in traditional medicine, a stable chemical composition in the plants used is essential. The variation in phytochemicals in the herbs, along with the criteria for plant collection, cultivation, and storage, presence of pesticides or toxins, makes the standardization of the quality of herbal products extremely difficult. The progression of phytochemical and phytopharmacological sciences has facilitated a comprehensive understanding of both the composition and biological activities inherent in numerous medicinal plant products. A trend has been observed in the growth of the value in the market of traditional medicine plants and products derived from them. The worldwide market for herbal medicine holds promising prospects and is likely to attain a value of USD 600 billion by 2033, the specific market for herbs in the form of the complete plant or powdered material is expected to reach USD 7.93 billion in 2027, and the market for capsules, tablets, and extracts is expected to be USD 117 billion by 2029 [56]. Despite the majority of research on medicinal herbs being focused on clinical therapy, it is noteworthy that there is an increasing trend in using them for the development of environmentally friendly methods for synthesizing nanomaterials. Primarily, by integrating botanical sources as the primary components in nanoparticle synthesis, the concerns associated with greenhouse gas emissions could be effectively mitigated.

## 5. Characteristics of Medicinal Plant Extracts to Synthesize NPs

Medicinal plants form an extensive repository of natural organic compounds, offering vast potential for nanotechnology applications. Plant extracts from leaves, flowers, wood, stem, roots, fruit peels, and pulps have been utilized for the green synthesis of NPs since they contain diverse phytochemicals such as polyphenols (flavonoids, phenols, phenolic acids, and terpenoids), alkaloids, and organic acids, among others (Figure 1). These compounds are known as antioxidants because of their ability to donate hydrogen or electrons as well as their stable radical intermediates [26,57,58,59]. These plant-derived compounds have a variety of functional groups, including hydroxyl (-OH), nitrile (-CN), aldehyde (-CHO), amines (-NH2), and carboxylic acid (-COOH), in their structures. These functional groups provide biocompounds with a redox capacity that allows them to participate in the biosynthesis of NPs as reducing, covering, chelating/capping, and preservative agents for NPs [32,60,61,62]. Despite knowledge of the redox mechanisms of these functional groups, there is still no fully elucidated mechanism for the synthesis of NPs. Due to the great variety and richness of biomolecules that may be part of plant extracts, there is still no clearly elucidated which plant-derived compound is specifically responsible for the reducing, oxidative, capping, or stabilizing properties. Variables that interfere in the CuNP synthesis process such as temperature, oxidation-reduction phenomena, agitation, and the participation of plant extracts are discussed in greater detail in Section 7.

Beyond only considering phytochemicals as an active ingredient in NP synthesis, it is relevant to consider that their antioxidant properties are related to multiple benefits of human health. Several plant biocompounds have already been found and reported to have a wide range of antifungal, antiviral, antibacterial, antiparasitic, cardioprotective, and anti-inflammatory actions [63,64,65]. The appeal of plant-derived medicines lies in their inherent guarantee of safety and absence of side effects. For example, phenols, flavonoids, quinones, tannins, coumarins, and phenolic acids play important roles as chemical defenses against insects and microorganisms. Remarkably, these compounds preserve their properties even when extracted from the plant [65]. There is evidence of a significant correlation between phenolic content and antioxidant activity. Many phenolic compounds exhibit antiviral, antiobesity, antidiabetic, and anti-inflammatory properties, as well as anticancer effects, due to their antioxidant nature [66,67]. Natural carboxylic acids, especially cinnamic acids, *in vivo* and *in vitro* induce apoptosis of colon and cervical cancer cells [68]. In recent times, there has been a persistent resurgence in the exploration of therapeutic plants for drug development. Hydroxamic acid and compounds with a benzamide group have potential as anticancer medication [69].

It has been observed that NPs synthesized with extracts of medicinal plants have better properties than NPs synthesized by another method [70]. A significant diversity of studies involving the exploration of plant bioactive compounds applied to the synthesis of NPs has indicated an extremely promising scenario, since it has been shown that the synthesized NPs acquire the antiviral, antibacterial, anti-insect, cardioprotective, anti-inflammatory, derived, and associated biomolecules that participate in their synthesis leading to multiple applications regarding the use of these NPs for a variety of applications.

The concentration, complex biochemical composition and quality of bioactive compounds present in plant extracts are variables that directly affect the efficiency of the synthesis process, as well as the stability, dimensions, and geometry of the NPs [60]. Hence, it is crucial to consider the extraction method for acquiring bioactive compounds from plants and to thoroughly characterize them. This becomes significant in the pursuit of discovering novel biomolecules intended for utilization in NP synthesis.

Another variable of consideration is the type of tissue from which the bioactive compounds will be obtained, since different plant components possess varying levels and distributions of the desired metabolites [31,66,71,72]. Extraction efficiency can vary significantly even if the same extraction variables are used—solvent, temperature, time, and methodology—but using different types of tissue [73]. For example, among the cases of the use of different tissues to obtain bioactive components that could participate in the synthesis of CuNPs is the use of flowers, leaves, and stems of *Gnidia glauca*. In this study, the extraction methodology was the same for all tissues and developed under the same conditions of temperature, aqueous solvent, drying and spraying pretreatment, and storage. No apparent change was observed in the efficiency of nanoparticle synthesis, and a similar pattern in the enhancement of the spectral intensity was observed in all the cases where CuNPs were synthesized using flower extracts, leaves, or steams. However, when analyzing the size of the synthesized CuNPs, different dimensions and shapes were observed. The CuNPs synthesized with flower extract had a dimension of 5 nm and a spherical shape, while those synthesized with leaf extract reached sizes of 70–93 nm with a spherical shape, but with irregular edges. On the other hand, the CuNPs synthesized using the stem extract presented a monodispersed distribution and were discretely placed without any sign of aggregation or agglomeration, indicating high stability. When the compositions of the three extracts were analyzed by means of FTIR, all extracts showed similar characteristic peaks, but with variation in their intensity, indicating the presence of similar functional groups, but with different proportions of the concentration of their bioactive components [74].

Due to the multifarious variables inherent in the nanoparticle synthesis process, a wide variety of shapes, sizes, and characteristics can be attained (Figure 3). Notably, the composition of the extract plays a vital role in determining the final characteristics of the NPs. In the section “Mechanism of CuNP biosynthesis,” we shall delve into the intricate mechanisms governing the formation of these NPs.

## 6. Extraction Methods to Obtain Plant-Derived Compounds

The development of methodologies for the extraction of bioactive compounds from plants has been carried out empirically. This does not mean that it does not have scientific and methodological support, but rather that there is a wide variety of solvents that can be used to promote the extraction of the compounds from the tissue into the solvent itself. Various combinations of time and temperature in the extraction process can be applied with the aim of achieving a differential extraction between the compounds of interest and other coextracted components based on their acidity, polarity, or molecular size [75,76].

Prior to the extraction process, the preparation of the plant material to be treated must be carried out. Conventionally, four stages are developed that go from the selection of the material, washing and disinfection, drying, and shredding or grinding. Shredding and grinding refer to processes used on the plant material prior to extraction. The degree of extraction is dependent on the particle size, with smaller particles resulting in a larger surface area for contact between the plant material and solvents, leading to a higher level of extraction [77].

Conventionally, bioactive compounds (phenolic compounds, organic acids, carotenoids, flavonoids, among others) are traditionally extracted by steam distillation, maceration, infusion, solvent extraction, Soxhlet extraction, pressing method, etc. [78]. Nonetheless, these methodologies have some disadvantages, such as long time requirements, higher solvent consumption, use of organic solvents that are not biocompatible for therapeutic applications or contact with humans or animals, degradation of thermolabile bioactive compounds, generation of polluting or toxic effluents, and in many cases low extraction yields [79,80,81].

Researchers have put considerable effort into finding efficient extraction methods in order to obtain a high extraction yield that allows the recovery of compounds with preserved and enhanced bioactivity. Among the advanced extraction methods that can be considered are modern Soxhlet, sonication, SFE (supercritical fluid extraction), PLE (pressurized liquid extraction), PHWE (pressurized hot water extraction), shockwave and MAE (microwave assisted extraction) developed in its two configurations, closed-vessel systems, and open-vessel systems [76,82]. These methods present diverse fundamentals that involve operational variables that can lead toward automation compatible with industrial and large-scale processes. On the other hand, it has been shown that the consumption of the vegetable sample and the solvent used is lower with the acquisition of bioactive extracts and more concentrated than those obtained by traditional methods and in a shorter operating time. It has also been reported that the SFE and MAE methods are compatible with the recovery of volatile and thermolabile compounds, compounds that are usually degraded in conventional methods [76,80,81]. These technologies are characterized by minimizing or removing highly toxic solvents and maintaining the natural environment. Therefore, they are considered sustainable and environmentally friendly.

The impact of different types of solvents on the extraction yield of biocompounds from medicinal plants has been analyzed. A determining factor in the choice of solvents is their polarity, since this must be compatible and similar to that of the interest compound. For example, hexane is more polar than acetone and is in turn more polar than methanol and ethanol, and finally, we have water as a less polar solvent [73]. However, the effectiveness of the solvent will depend on the type and diversity of compounds to be extracted: ethanol may be more effective for the extraction of flavonoids, while methanol is more suitable for the extraction of phenolic compounds, or a mixture of both solvents may be required, and this may vary depending on the plant source in question [65,83].

The biological activities of the plant-derived compounds are subject to the quality and proportion of the active principles in the different vegetable preparations. A recent work analyzed the phytochemical profile (antioxidant activity, total polyphenol content, and flavonoid content) of 12 medicinal plants treated through three different extraction methods: common solvent extraction, ultrasonic-assisted extraction, and microwave extraction. The experimental results highlighted that the type and degree of active biocompounds in the analyzed extracts differed according to species, the processing procedure previous extraction, and the extraction method. Processing as a powder leads to a higher content of active ingredients compared with processing as crushed. The comparative study of the three extraction methods led to the conclusion that the optimal subtraction procedure for the subtraction of active ingredients from most medicinal plants was the microwave procedure, followed by ultrasonic-assisted extraction, and finally the common solvent extraction method [77].

Despite the great variety of extraction procedures, solvent options, equipment designs, and the development of more advanced methods with improved extraction efficiency and effectiveness, the predominant method linked with CuNP synthesis involves heating the extraction in water within a temperature range of 30 °C to 100 °C. This approach has been consistently applied across the works listed in Table 1 of this review. It would be very interesting to evaluate whether the properties of the NPs are also influenced by the extraction method, potentially leading to variations in the concentration and distinct composition of biomolecules present in these plant extracts. The biological activities of the plant-derived compounds are subject to the quality and proportion of the active principles in the different botanical preparations.

## 7. Mechanism of CuNP Biosynthesis with Plant Extracts

Although the specific mechanism of nanoparticle biosynthesis with plant extracts has not been fully elucidated, different potential mechanisms have been proposed. One of the most widely accepted models suggests that particle formation is initiated by a rate-limiting nucleation step followed by a growth or extension phase [1,6,26,84]. The classic nucleation theory posits that the rate-limiting monomer aggregation (i.e., the assembly of a certain number of individual molecules) is required for the formation of a primary, critical nucleus [1,84,85]. In the context of nanoparticle synthesis, the reduced metallic ions will stochastically interact with each other and form ordered arrangements resembling a crystalline phase (Figure 2).

Remarkably, the free energy per monomer in the solution is higher than the one in the crystalline phase. Consequently, individual monomeric molecules demonstrate a propensity to aggregate so a lower-energy state can be achieved. Small nuclei with high surface areas will combine to decrease the system’s free energy. Once a nucleus of an appropriate size arises, it will inevitably grow and give rise to the new macroscopic phase, ultimately facilitating particle growth [1,84,85].

The conditions in which nanoparticle biosynthesis is performed have a direct effect on nucleation and particle growth kinetics [1]. Factors such as reaction conditions (agitation speed, temperature, pH, reaction time), the choice of reducing agent, the nature and concentration of the precursor salt, and particularly the biological extract collectively dictate the characteristics of the synthesized NPs. While the optimization of these variables for nanoparticle biosynthesis may present a complex challenge, it also affords the opportunity to readily modulate and fine-tune the properties of the NPs until they align precisely with the desired specifications.

Various experimental protocols have been established to facilitate the previously described mechanism for nanoparticle synthesis (Figure 2). The copper precursor is mixed with the herb extract to achieve the desired copper concentration. A wide range of precursor concentrations have been reported in the literature. For instance, copper chloride has been studied within a concentration range of 10 to 250 mM, copper nitrate from 0.1 to 100 mM, copper acetate from 3 to 500 mM, and copper sulfate from 1 to 1000 mM [86]. Subsequently, the reaction mixture is stirred to ensure thorough mixing with the metal precursor. It is then subjected to vigorous stirring at a determined temperature for several hours.

There is no universally applicable temperature that suits all types of plant extracts; rather, it is a variable that requires standardization (Figure 4) [87]. It has been observed that at the same temperature, different extracts can result in NPs of different sizes. For example, protocols operating at 27 °C produced NPs with sizes ranging from 80 to 110 nm when using *Ixiro coccinea* leaf extract [88], while the utilization of *Cissus vitiginea* leaf extract resulted in NPs with sizes ranging from 5 to 20 nm [89]. Typically, green synthesis is performed at room temperature (RT) or higher temperatures up to 120 °C, for which an oil bath is often used. Hybrid protocols have also been reported, where the mixture is briefly held at a high temperature, ranging from minutes to a few hours, followed by a longer incubation period (more than 12 h) at room temperature and in the dark [59,90]. However, the order of each stage has also been reported in reverse order, first at room temperature followed by a high-temperature step [91]. There are even reports where an autoclaving procedure is used [92].

Regarding the participation of plant extracts in the synthesis of CuNPs, Fourier-transform infrared spectroscopy (FTIR) analysis shows that the reduction of metal ions, such as copper, to NPs, does not depend on a single biomolecule, in contrast to chemical synthesis. In green synthesis, various biomolecules like phenols, alkaloids, proteins, and organic acids are involved, depending on the composition of the selected plants [13,93]. Chandraker et al. conducted a study in 2020 on the phytochemical analysis of the extract of the leaves of *Ageratum houstonianum* Mill. FTIR analysis demonstrated that this group of compounds plays a pivotal role in the synthesis of CuNPs [94].

Notably, studies involving plant extracts do not definitively pinpoint the specific biomolecule responsible for NP synthesis and stabilization due to the process’s inherent complexity. Nevertheless, several investigations have highlighted molecules such as ascorbic acid, which display dual roles as reducing and stabilizing agents in the synthesis of gold and silver NPs [95], as well as copper NPs [96]. Furthermore, these effects have been observed with copper, silver, and bimetal NPs [97].

These results demonstrate that biomolecules with antioxidant capacity present in plant extracts are responsible for the formation of NPs. More detailed studies where these compounds are individually tested will likely facilitate a deeper understanding of their role in NP synthesis.

## 8. Downstream Process of NP Synthesis

Once the NP formation reaction is complete, a brown–black product is observed. The solid product is then separated for subsequent analysis and use (Figure 3). NPs can be isolated from the reaction mixture through filtration and high-speed centrifugation [98]. Both procedures, centrifugation and filtration, present different configurations, which could represent advantages in the recovery of NPs. There are methods that can combine both procedures, for example, filtration devices fitted in centrifugation units, which could separate possible solid NPs resulting from the synthesis or lead to a selective separation of various groups of NPs through the correct choice of the limit of molecular weight (MWCO). However, these basic separation processes must be carried out at speed conditions that avoid the compaction of the NPs by the driving forces of the separation (driving force of the filter, such as gravity—g-force or centripetal force). In other words, the process of recovery, purification and conditioning of the NPs (downstream process), could lead to deformation, caking, and aggregation of the synthesized NPs. These effects may result in challenges during NP characterization and undesirable characteristics could be observed resulting from the downstream process and not necessarily from the synthesis process [99,100].

Other reports use dialysis membranes with MWCO as a straightforward purification method. This method allows small organic molecules to pass through the membrane while retaining surface passivating agents that are linked to NPs within the dialysis membrane. However, it is worth noting that this approach can be time-consuming [101].

Additionally, it is important to consider other post-synthesis heat treatments like calcination and annealing, which are commonly used for NP decomposition and purification (Figure 3). Lyophilization has been used for the conservation and stability of various biomolecules applied in the biopharmaceutical field, so it is not uncommon for it to have been identified as an operational alternative for the long-term stabilization of NPs. Leaving the synthesized NPs in aqueous media limits their application, since low stability has been observed under these conditions, as well as the propagation of microorganisms [102,103].

Finally, the synthesized NPs must be evaluated to know their main characteristics. The main techniques used for this purpose are mentioned below. UV-visible spectroscopy helps to confirm the bioreduction of the nanoparticle. Fourier-transform infrared spectroscopy (FTIR) is used for the identification of biomolecules and a functional group involved in the reduction and stabilization of NPs. Transmission electron microscopy (TEM) and scanning electron microscopy (SEM) analysis are conducted to study the size, shape, and morphology of the nanoparticle. X-ray diffraction analysis (XRD) analysis is used to find the crystalline structure of the nanoparticle. Dynamic light scattering (DLS) has been used to measure the size and size distribution profile of NPs.

## 9. Relationship of the Synthesis Process Variables and CuNP Properties

To evaluate the relationship between various conditions of the steps used to obtain NPs and the size achieved, we compared different reports. Table 1 presents a compilation of cases of CuNPs synthesized by plant extracts from various medicinal plants. These CuNPs have exhibited relevant functional properties in various fields of application. This table also specifies the extraction procedure applied to obtain the reducing agents from specific plant parts used, as well as the precursor copper salt for the synthesis of NPs and a brief description of the green synthesis process. Additionally, it specifies the potential application of the synthesized CuNPs. Through the data collected from various reports (listed and described in Table 1), Figure 4 was constructed, in which the obtained diameters of the NPs synthesized through complex plant extracts have been represented. Figure 4 shows that there is no trend that indicates the dependence of the NP diameter on the variables of temperature or synthesis time applied. We believe that these large variations could be correlated to the complexity of the mixture of phytocompounds present in the plant extracts, even more so when there are reports in which whole plants or combinations of plant extracts were used. In some cases, the obtaining of NPs of sizes between 6 and 20 nm was observed, with no report of increased dimensions or aggregation. These reports correspond to NPs synthesized from the plant extracts of *Krameria* sp. in conjugation with (CuSO_4_)·5H_2_O, exhibiting a diameter of 6 nm without apparent aggregation [104]. A similar case was found in NPs obtained with extracts of *Galeopsis herba* and *Carum carvi* in conjugation with the same precursor salt Cu(NO_3_)2 ·3H_2_O, where the synthesized NPs had a diameter of 5–10 and 12 nm, respectively [90,105]. However, the synthesis conditions of the three green synthesis protocols do not present any convergence in the applied variables.

Although the diameter or size of the NPs is not the only criterion to be evaluated, since other functional properties must also be considered, it is a parameter that can give us a reference of stability as it is related to structural changes or aggregation phenomena. The size has been related to the mechanisms of functionality, for example, in the antidiabetic potential, evaluated through inhibition of amylases or lipases, the authors report that this inhibitory effect is due to the fact that the NPs physically interfere in the structure of the protein [74]. It has also been described that the antiviral and bactericidal activity present a correlation between the size of the NPs and effectiveness [106,107].

Regarding morphology, the most recurrently observed in CuNPs is spherical, with the exception of some reports, such as that of NPs synthesized from extracts of *Hagenia abyssinica* leaves, in which they indicated spherical, hexagonal, triangular cylindrical, and prismatic shapes [108]. Particularly, the synthesis of CuNPs from extracts of *Falcaria vulgaris* presented multiple functionalities: antioxidant, antifungal and antibacterial activity, in addition to presenting cutaneous wound healing potential without any cytotoxicity. In this work, relatively conventional extraction and synthesis methods are reported, with the addition of NaOH to equilibrate the pH during the reaction, leading to the obtaining of CuNPs with a 20 nm diameter and spherical shape [109].

There are few reports that address an analysis of the stability of functionality or conservation of initial characteristics of the NPs synthesized by a green pathway [110,111,112]. In most of the reported works, a robust approach is made towards the synthesis methods, the characterization of the NPs synthesized with plant extracts, and their potential applications. However, the mention of monitoring its stability over a long period of time is barely addressed or not mentioned at all.

**Table 1 micromachines-14-01882-t001:** Examples of biosynthesized CuNPs by medicinal plant extracts.

Plant	Extraction Procedure	Extract Phytochemical Compounds	Precursor	Synthesis Method	NPs Characteristics	Applications	Ref.
*Azadirachta indica*, *Hibiscus rosa-sinensis*, *Murraya koenigii*, *Moringaoleifera*, and *Tamarindus indica*	Powdered leaves were boiled with distilled water for 20 min at 60 °C. Cooled at RT and filtered.	Alkaloids, carbohydrates, flavonoids, glycosides, phenolic compounds, saponins, steroids, tannins, and volatile oils.	CuO	Mix of plant extract and precursor was boiled at 80 °C until the formation of a deep-green paste. The paste was heated at 400 °C for 2 h resulting in black colored powder.	Spherical with particle size range between 9.8 and 10.77 nm.	Antioxidant activity and cytotoxicity against four cancer cell lines such as human breast (MCF-7), S cervical (HeLa), epithelioma (Hep-2), and lung (A549).	[113]
*Kigelia africana*	Fruit extract was obtained by ethanol without thermal treatment for 48 h with light protection.	Alkaloids, anthraquinone, flavonoids, glycosides, phenols, quinones, saponins, steroids, tannins and terpenoids.	Cu(CH _3_COO)_2_	Mix of plant extract and precursor was stirred for 3 h and the absence of light for 24 h The mixture was centrifuged and the precipitate was washed and dried at 80 °C.	The study does not report the morphology or size of NPs, it focuses only on antimicrobial activity.	Antimicrobial activity against *Escherichia coli*, *Pseudomonas aeruginosa*, *Salmonella typhi*, *Shigella* sp., and *Staphylococcus aureus.*	[114]
*Centella asiatica*	Powdered leaves were infused in double distilled water at 80 °C for 30 min. Cooled at 4 °C and filtered.	Alkaloids, flavonoids, saponins, terpenoids, tannins, glycosides, carbohydrates, quinines, organic acids, centellose, phellandrene, and vitamin C.	CuCl_2_∙2H_2_O MnO_2_	A mixture of MnO_2_ and plant extract was stirred for 20 min at room temperature. Following this, CuCl_2_ solution was added dropwise with vigorous stirring and heated for 4 h at 80 °C.	Cu/MnO2 nanocomposites, size range 10–30 nm.	High catalytic activity for the reduction of inorganic and organic dyes in aqueous media at ambient temperature. (Congo red, rhodamine B, and methylene blue).	[91]
*Camelia sinensis*	Powdered leaves were infused at 75–85 °C for 30 min and continuously stirred. The extract was centrifuged and the supernatant was separated for reaction.	Polyphenols, flavonoids, and alkaloids.	CuCl_2_	Mix of plant extract and precursor salt was heated at 75–85 °C for 1 h and continuously stirred. The mixture was cooled at RT for 2 h and then centrifuged.	Agglomerated form with an average size of 60 ± 6 nm.	Efficient photocatalyst in dye degradation (using bromophenol blue).	[61]
*Ageratum houstonianum*	Fresh leaves were washed with water, then chopped and boiled at 60 °C for 20 min and filtered.	Alkaloids, flavonoids, tannins, triterpenes, diterpenes, steroids, and saponins.	CuCl_2_	3 mM solution of CuCl_2_ as precursor was stirred for 2 h; then mixed with leaf extract and stirred for another 24 h at room temperature.	Size around 80 nm, agglomerate, and not specific shape. NPs behave as a semiconductor.	Dye degradation against Congo red (azo dye). Antibacterial activity against *E. coli* (MTCC no. 40).	[94]
*Ehretia acuminata*	The fruit, leaves, and bark of *E. acuminata* were dried, ground, soaked and macerated with dichloromethane or methanol for 14 days.	Phenolic acids, steroids, terpenoids, polyphenolic compounds, tannins and flavonoids.	CuCl_2_·2H_2_O	L-ascorbic acid and precursor solution were mixed and heated at 100 °C continuously until the color changed (20 h).	NPs of 500 nm. The shape was not reported. Green NPs and phytochemicals were coated on a cotton textile surface.	Antiviral action was shown by the fabrics treated with CuNPs tested by coronavirus-infected Vero-E6 cultures.	[115]
*Aloe vera*	Powdered leaves were boiled for 5 min at 80 °C with deionized water.	Polysaccharides, flavonoids, and phenolic compounds.	Cu(NO_3_)_2_·3H_2_O	Mix of plant extract and precursor was stirred for 24 h at 100–120 °C.	Monoclinic phase with average particle size of 20 nm.	Bactericidal properties against three fish bacterial pathogens: *Aeromonas hydrophila*, *Pseudomonas fluorescens* and *Flavobacterium branchiophilum.*	[116]
*Galeopsis herba*	Powdered *Galeopsidis herba* was mixed with water and stirred for 50 min at 85 °C and filtered.	Iridoids, saponins, flavonoids, phenolic acids and tannins.	Cu(NO_3_)_2_·3H_2_O	The extract was mixed with Cu(NO_3_)_2_ in 90:10 (W:W) proportion and stirred 4 h at 80 °C then was stored for 24 h in dark place at 25 °C.	The size of NPs was 5–10 nm with spherical shape, dispersed and crystalline.	CuO- NPs showed high antioxidant activity against free radicals with a value of 4.12 µg/mL. NPs presented catalytic activity.	[90]
*Hagenia abyssinica*	Powdered leaves were boiled in deionized water at 50 °C for 1 h, with light protection.	Tannins, anthraquinone glycosides, cardiac glycosides, phenolic compounds.	Cu(NO_3_)_2_ ·3H_2_O	Mix of plant extract and precursor salt has been incubated at RTfor 24 h. The precipitate was washed and dried.	Spherical, hexagonal, triangular and cylindrical, and prismatic shapes. Size range of 10–50 nm.	Antibacterial activities against *Escherichia coli Pseudomonas aeruginosa*, *Staphylococcus aureus*, and *Bacillus subtilis.* Cu NPs presented concentric circular patterns with d-spacing of 0.24 nm.	[108]
*Cinnamomum zelanicum*	Powdered leaves of were macerated in boiling water for 6 h. The extract was filtered and evaporated to concentrate.	Cinnamic, coumaric, sinapic, ferulic and caffeic acids, camphor, linalool, benzyl benzoate, cinnamyl acetate, eugenol, and cinnamaldehyde.	Cu(NO_3_)_2_·3H_2_O	A mixture of plant extract and precursor salt was heated and stirred at 65 °C for 24 h. CuNPs were washed, centrifuged and dried at 55 °C.	Spherical morphology with size of 19.55 to 69.70 nm.	Antioxidant activity and anti-lung carcinoma properties against NCI-H2126, NCI-H1437, NCI-H1573, and NCI-H661 cell lines.	[117]
*Berberis vulgaris*	Leaves were macerated in water for 3 h at 90 °C. The extract was filtered and evaporated to concentrate.	Carbohydrates, fiber, several minerals, berberine, vitamin C, iron, zinc, copper and anthocyanins.	Cu(NO_3_)_2_·3H_2_O	Plant extract and precursor salt mixture was heated and stirred at 65 °C for 24 h.	Spherical with size of 10–100 nm.	Cardioprotective potential against isoproterenol-induced myocardial ischemia in mice.	[118]
*Carum carvi*	Leaves were macerated in double-distilled water using shaker incubator for 24 h at 45 °C, then cooled at RT and filtered.	Carvacrol, carvone, α-pinene, limonene, γ-terpinene, linalool, carvenone, and *p*-cymene, carveol, camphene and fenchen.	Cu(NO_3_)_2_·3H_2_O	Plant extract was added dropwise to precursor solution, with continuous stirring until color change after 12 h at 45 °C.	Regular and homogenous distribution spherical form with 12.4 nm size.	CuONPs had positive effect on the various physiological and biochemical characteristics of the *Solanum lycopersicum* seedlings (increased sugar content and pigment).	[105]
*Syzygiumalternifolium*	Dried fruits were boiled at a water bath at 80 °C for 30 min. The extract was filtered and stored at 4 °C.	Alkaloids, anthocyanins, anthraquinones, glycosides, emorins, flavonoids, and phenols.	CuSO_4_·5H_2_O	Plant extract was mixed with salt precursor at 50 °C for 2 h. The pH was adjusted to 8.2–9.0 by adding NaOH.	Spherical shape, size from 2 to 69 nm, non-agglomerated and polydisperse nature.	Antiviral ability against Newcastle disease virus.	[119]
*Falcaria vulgaris*	Powdered leaves were infused under magnetic stirring for 30 min at 50 °C.	Carvacrol, spathulenol, genistin, rutin, quercetin-3-O-glucoside, and quercetin.	CuSO_4_·5H_2_O	Mix of plant extract and precursor salt was rapidly stirred. Then NaOH was added to catalyzed and adjusted to pH 12. Stirring continued for 1 h.	Spherical shape with average diameter size of 20 nm.	Antioxidant activity. Antifungal activity against *C. albicans*, *C. glabrata C. guilliermondii C. kruse*. Antibacterial activity against *E. coli* and *S. aureus.* Cutaneous wound healing potential without any cytotoxicity.	[109]
*Orobanche aegyptiaca*	Powdered stems were treated by reflux extraction with distilled water, for 30 min.	Polyphenols, tannins, alkaloids and peptides.	CuSO_4_·5H_2_O	Mix of plant extract and precursor salt was stirred for 10–15 min at room temperature. The resultant mixture was kept in dark for 72 h.	Spherical shape with particle size less than 50 nm.	Nematicidal properties against *Meloidogyne incognita.* Antibacterial activity against *Escherichia coli* and *Staphylococcus aureus.*	[120]
*Gnidia glauca* *Plumbago zeylanica*.	Flower, leaves, and stem of *G. glauca* and leaves of *P. zeylanica* were boiled at 100 °C for 5 min. Extract was filtered, and stored at 4 °C.	*G. glauca:* Heterocyclic polyol components, flavonoids, and terpenoids. *P. zeylanica:* phenolics, flavonoids, reducing sugar, citric acid, and plumbagin.	CuSO_4_·5H_2_O	Mix of each plant extracts and precursor salt was stirred within 5 h at 100 °C.	Variable size according to plant extract used from 5–93 nm. Irregular brush border rods and spherical shape.	Antidiabetic activity evaluated by *α*-amylase inhibitory assay using the chromogenic 3,5-dinitrosalicylic acid (DNSA) method.	[74]
*Eucalyptus camaldulensis*, *Azadirachta indica*, *Murraya koenigii*, *Rosa rubiginosa* and *Datura stramonium*	Extract from leaves of each plant were obtained by soaking individually in aqueous ethanol (80% *v*/*v*) at RT for 3 h.	Alkaloids, flavonoids, terpenoids, polyphenols and proteins.	CuSO_4_·5H_2_O	Mix of plant extract and precursor salt was stirred at 80 °C for 10 min. After, mix was continuously stirred at 200 rpm for 24 h at RT and centrifuged.	Spherical shape with Variable size according to plant extract used from 41–65 nm.	Showed destruction of cell membrane and cell lysis of *S. aureus*, *S.* *mutans*, *E. coli*, *K. pneumoniae and S. typhi* and the multidrug-resistant *P. aeruginosa.*	[59]
Prunus *nepalensis*	Fruit extract was obtained by heated in deionized water in a water bath to 80 °C for 1 h.	Polyphenolic compounds, flavonoids, amino acids, alkaloids, saccharides, and tannins.	CuSO_4_·H_2_O	Mix of plant extract and precursor salt was stirred and incubated at RT overnight under dark conditions. Then the precipitate was centrifuged and washed.	Spherical with size ranging from 35 to 50 nm.	Anticancer activity on human breast cancer cell lines by increasing the gene expression of apoptotic genes in a dose-dependent manner.	[121]
*Nigella sativa*	Seeds were heated in water at 30 °C for 40 min. The extract was cooled, filtered and centrifuged.	Enzymes, phenols, flavonoids, terpenoids.	(CuSO_4_)·5H_2_O	A solution of precursor salt was heated up to 80 °C on a hot plate and seed extract was added dropwise with constant stirring at 150 rpm.	NPs with size of 98.23 nm with changes of size particle increase CuNPs concentration. Form was not reported.	Antiobesity activity tested by lipase and amylase inhibition assays. High antibacterial activity against *Pseudomonas aeruginosa* and *E. coli.*	[122]
*Zingiber officinale*	Ginger root powder was boiled at 50–60 °C for 10 min. Extract was filtered, and stored at 4 °C.	Polyphenols, such as 6-gingerol, 8-gingerol, and 10-gingerol.	CuSO_4_·5H_2_O	Mix of plant extract and precursor was stirred at RT until color changed. After the solution was centrifuged and the precipitate was dried and heated at 90 °C for 12 h.	Crystalline configuration with size of 60 nm.	Antibacterial activity against *Staphylococcus aureus* and *Escherichia coli.*	[123]
*Haplophyllumtuberculatum*	Complete dried plant was placed in a water bath at 70 °C with continuous stirring, for 3 h. After that, it was left at 4 °C and it was valid for use for a week.	Gallic acid, ferulic acid, catechin, quinol, syringic, caffeic, vanillic, ellagic and cinnamic acids, catechol and benzoic acid.	Cu(NO_3_)_2_·3H_2_O	Mix of plant extract was stirred and the precursor was added slowly at 500 rpm at RT. The solution was centrifuged and the precipitate was dried (18 h at 50 °C).	Amorphous particles with the average of about 85 nm.	Nematicide activity against *Meloidogyne incognita*.	[124]
*Krameria* sp.	*Krameria* roots were macerated and boiled.	Tannins (cate-chins and proanthocyanidins), rhataniatannic acid, and tannic acid.	(CuSO_4_)·5H_2_O	Mix of plant extract and precursor salt was stirred at 70 °C during 3 h. After that, the solution was centrifuged and the precipitate was rinsed and dried at 80 °C for 6 h.	Spherical NP’s and average size of 6.16 nm.	Antioxidant therapy. Antimicrobial agent against *Escherichia coli*, *Staphylococcus aureus*, *Alternaria alternata*, and *Fusarium oxyporium.*	[104]
*Nigella sativa*	Seeds were heated in water at 30 °C for 40 min. The extract was cooled, filtered and centrifuged.	Enzymes, phenols, flavonoids, terpenoids.	(CuSO_4_)·5H_2_O	A solution of precursor salt was heated up to 80 °C on a hot plate and seed extract was added dropwise with constant stirring at 150 rpm.	NPs with size of 98.23 nm with changes of size particle increase CuNPs concentration. Form was not reported.	Antiobesity activity tested by lipase and amylase inhibition assays. High antibacterial activity against *Pseudomonas aeruginosa* and *E. coli.*	[122]

## 10. Application of Biosynthesized CuNPs

Different CuNP properties have been described, such as bactericidal, antifungal, nematocidal, antidiabetic, antioxidant, anticancer, antiobesity, cardioprotective, healing activity, and catalytic reduction and degradation of dyes. CuNPs can be applied for medical treatments because of their chemical, physical, and biological activities; several authors have reported potential applications of these materials in the nanomedicine field. Antibacterial, antioxidant, catalytic, and other properties must be evaluated against diverse diseases. One of the most important advantages of CuNPs obtained by green synthesis is that their precursors make them non-toxic for humans, affordable, and easily optimized [87]. Because of their capacity to interact with biological systems, these NPs can be used as antioxidants, anticancer, and antimicrobial agents [87,104,121].

### 10.1. Therapeutics

Certain plants have been identified for their remarkable antitumor properties, alongside other advantageous traits. Different plants have been used as biological sources for the synthesis of CuNPs and they have shown cytotoxicity for multiple types of cancer in a dose-dependent manner without affecting healthy cells. Multiple studies have been reported with cell lines of breast cancer, cervical cancer, epithelioma, hepatocellular carcinoma, lung carcinoma, colorectal cancer, ovarian cancer, prostate cancer, endometrial cancer, and melanoma, among others [86]. A study involving the use of copper oxide nanoparticles (CuONPs) derived from *Azadirachta indica*, *Hibiscus rosa-sinensis*, *Murraya koenigii*, and *Moringa oleifera* demonstrated antioxidant and cytotoxic effects when assessed against diverse cancer cell lines [113]. These findings highlight the promising role of these plant-derived NPs in cancer research and therapeutic applications.

Among the cancer treatment options, such as surgery, radiotherapy, and chemotherapy, selective cancer photothermal therapy (PTT) is a promising modality useful to prevent the recurrence of cancer. PTT involves delivering focused near-infrared (NIR) laser energy to the tumor, generating high heat, and offering several advantages such as minimal invasiveness, absence of drug resistance, low toxicity, and minimal side effects. A new alternative of NPs as a photothermal agent, with a promising ablation effect on 4T1 cells when exposed to an 808 NIR laser, was obtained through the synthesis of copper-doped carbon dots (Cu-dCD) using *Alcea* sp. leaf extracts as the organic precursor and CuSO_4_ [125].

Another area of great interest in therapeutic medicine is the treatment of chronic wounds. CuNPs exhibit remarkable healing properties by facilitating the release of growth factors that aid in the anti-inflammatory process of wounds and substantially enhance antibacterial and antioxidant activities. Moreover, copper demonstrates superior biocompatibility compared to other metallic ions, promoting regeneration and enhancing the quality of the skin [18].

### 10.2. Metabolic Disease Treatment

Obesity and diabetes are serious, chronic medical conditions associated with a wide range of life-threatening conditions. Plant extracts possess antioxidant, anti-inflammatory, and insulin-sensitizing characteristics and are considered a possible choice for the treatment of metabolic disorders due to their low risk of side effects. Phytonanomedicine accelerates the development and use of technologies or materials that interact with the body at a molecular level with a high level of specificity, which could be an optimal approach to combat metabolic diseases [126,127]. Likewise, the discovery of an improved drug delivery system has led to the formation of green biosynthesized NPs, which have increased the bioavailability, biodistribution, solubility, and balance of plant-derived products [128]. Considering that metabolic diseases are a new field of application for green NPs, in the period from 2017 to 2023 there were few reports of different conjugations of metals and plant extracts. However, for the most part, it is possible to appreciate a trend towards the use of gold precursor salts for the synthesis of NPs. As well as the antiobesity, antidiabetes, and cardioprotective properties, the main pharmacological roles were explored.

AuNPs synthesized from *Dendropanax morbifera Léveille* extract display antiadipogenic properties in 3T3-L1 and HepG2 cell cultures stimulated with cocktail media to generate obese and fatty liver disease models. Using real-time PCR, the expression of the adipogenic genes PPARγ, CEBPα, Jak2, STAT3, and ap2, as well as the hepatogenic genes PPARα, FAS, and ACC, were evaluated, their expression levels being associated with negative control of the adipogenesis process. The triglyceride content was also evaluated, and a substantial reduction was observed with the addition of AuNPs at the post-confluent stage [129].

The application of copper NPs in the treatment of metabolic diseases can be evidenced by fewer than a dozen publications, which leads to an extensive area of research to be developed in this regard.

CuNPs bio-generated using medicinal plant extract of flower, leaf, and stem of *G. glauca* and leaves of *P. zeylanica* inhibit porcine pancreatic α-amylase and α-glucosidase, which are key enzymes of carbohydrate metabolism. By circular dichroism analysis, authors found that CuNPs conduce to structural modification in both enzymes, theorizing that this conformational change was the cause of the inhibition mechanism [74].

CuNPs were synthesized using an extract of *Nigella sativa* seeds. These NPs presented antiobesity activity, evaluated through lipase and amylase enzyme inhibition assays. The application of CuNPs led to an approximate reduction of enzymatic activity of 30% in both cases. Amylase inhibition is associated with helping to decrease the digestion of glucose and carbohydrates, which targets to reduce postprandial hyperglycemia, and lipase inhibition helps to inhibit the dietary lipids conversion into fatty acids [122].

Another case related to diseases of a metabolic nature was in which the cardioprotective effects of CuNPs green-formulated by *Berberis vulgaris* leaf extract was verified. CuNP addition reduced the inflammatory milieu in the heart of mice with myocardial ischemia, thereby blocking the upregulation of proinflammatory cytokines (interleukin-1β (IL-1β), tumor necrosis component alpha (TNFα), and interleukin 6 (IL-6) [118].

### 10.3. Antibacterial Activity

The growing antibiotic resistance of bacteria has emerged as a significant concern in the field of human healthcare. The antimicrobial mechanisms of copper depend mainly on the physical form (ions or NPs) in which it is applied, followed by the oxidation state, concentration, form of application, and presence of other contaminants. CuNPs have demonstrated considerable efficacy in combating drug-resistant bacterial infections. Their high surface-to-volume ratio and positive charge enhance their affinity for the cellular membrane, altering the electrical potential difference of the cell and causing membrane depolarization and leakiness [19,130]. Additionally, the ability of NPs to simultaneously target multiple pathways to disrupt bacterial cells makes it challenging for bacteria to develop resistance against them. Another mechanism suggests that copper ions affect protein folding and stability, promoting protein aggregation under anaerobic conditions [131]. Biosynthesized CuNPs have also been shown to inhibit biofilm formation [132]. In addition, copper damages microbial cells by generating reactive oxygen species, ROS [133], and replacing or binding the native cofactors in metalloproteins [134].

CuNPs have emerged as a propitious therapeutic alternative for combating bacterial infections, offering a potential solution to address the escalating challenge of antibiotic resistance and diminishing reliance on conventional antibiotics to treat bacterial infections. Some examples of bacteria that show sensibility to CuNPs are *Staphylococcus aureus*, *Escherichia coli*, *Salmonella*, and *Pseudomonas aeruginosa*, among others [14]. Due to these antibacterial properties, multiple works have been developed to evaluate their efficacy against microorganisms that pose challenges across various sectors.

Copper ions have been used for crop protection; however, the accumulation of Cu2+ ions will be problematic in soils and become an environmental problem. Because of this, CuNPs are proposed as an option for agriculture due to their antibacterial and antifungal properties. Nanotechnology is a new tool for controlling diseases in agriculture [135,136]. In crop protection, the size of CuNPs determines efficacy against phytopathogens, and diverse studies demonstrate higher efficacy with smaller diameters and incorporation of metal NPs into biopolymer matrices like chitosan, showing enhanced activity against *Sclerotium rolfsii* and *Rhizoctonia solani*, *Fusarium* spp., and other pathogen fungi [135,137].

The effect of copper NPs as a control agent for phytopathogens has been attributed to their ability to cause damage to the cell membrane and disruption in the metabolic processes of pathogens [138,139]. The size and shape of the nanoparticle are determining factors; therefore, the optimization of the synthesis processes is the key factor in obtaining NPs with the desired characteristics according to the application.

Kaningini et al. in 2023 found that CuONPs synthesized with *Athrixia phylicoides* extracts can be an alternative for the treatment of bacterial diseases caused by *Bacillus cereus* and *Staphylococcus aureus.* In addition, these NPs were not toxic in human cells either [140].

CuNPs have shown the inhibition of the growth of bacteria found in the mouth that cause cavities and other dental problems. The growth of bacteria like *Aggregatibacter actinomycetemcomitans*, *Lactobacillus acidophilus*, and *S. mutans* has been observed [134,141]. For this reason, the incorporation of NPs into oral disease prevention medicines, prostheses, and tooth implantation has caught the attention of researchers.

The action mechanism of copper NPs continues to be studied; it is considered that it is closely related to the characteristics of the precursor. It has been reported that they can cause DNA breakage, promoting cell death; metal NPs can inhibit growth or infective capacity. The characteristics that define their antibacterial activity are related to the size, form, concentration, and sensitivity of the microorganism [142]. Majumdar et al. in 2019 reported that the activity of CuNPs against *Xanthomonas oryzae* depends on their size and concentration, leading to an increased production of reactive oxygen species [107].

Chatterjee et al. in 2014 revealed that the antibacterial activity of CuNPs is more complex than it appears. It involves various mechanisms and is not directly related to the release of copper ions, but rather to the formation of a reactive complex between the cell medium and CuONPs [143]. Their experiments with *E. coli* demonstrated that cell death results from the generation of reactive oxygen species (ROS), which cause protein oxidation, lipid peroxidation, and DNA degradation.

In copper NPs obtained by green synthesis with *Angelica keiskei* leaf extract, it was observed in transmission electron microscopy (TEM) analysis that CuNPs adhered to and subsequently broke the cell wall of Gram-positive and Gram-negative bacteria, causing their cell death [144].

The most reported antibacterial mechanisms are focused on silver NPs. In the case of copper NPs, there is still an area of opportunity for the development of more detailed studies in this regard.

### 10.4. Antiviral Activity

The evolutionary process of viruses is substantially fast and efficient, which leads to an emerging challenge worldwide. For this reason, it is vitally important to develop new and innovative strategies to combat them, maximize the usefulness of existing drugs, and explore other approaches that interrupt viral propagation processes.

Aqueous extracts of medicinal plants such as *Nepeta nepetella*, *Nepeta coerulea*, *Nepeta tuberosa*, *Dittrichia viscosa* and *Sanguisorba minor magnolii* showed a clear antiviral activity against two different DNA and RNA viruses [145]. Different solvents extract from *Plumeria alba*, *Ancistrocladus heyneanus*, *Bacopa monnieri*, *Anacardium occidentale*, *Cucurbita maxima*, *Simarouba glauca*, and *Embelia ribes* exerted anti-dengue virus and anti-chikungunya virus activities [146]. Flower extract from *Artemisia cina* showed promising antiviral activity against zoonotic highly pathogenic influenza virus [147]. *Ficus rubiginosa* could represent an interesting natural resource of antiviral compounds against HSV-1 and HCoV-229E [148]. The exploration of the conjugation plant extracts from these plants, such as those listed above with antiviral activity, in green nanoparticle synthesis processes would be of great interest to evaluate the transfer of antiviral capacity to the generated NPs, thus contributing to the generation of new potentiated antiviral alternatives.

Numerous efforts have been made to generate NPs with antiviral capacity. Among the various NPs biosynthesized with plant extracts, silver nanoparticles (AgNPs) have been successfully distinguished as antiviral agents against SARS-CoV-2 [149], HIV [150], chikungunya virus [151], and herpes virus [152], among others. Although the antiviral mechanism of biosynthesized SNPs has not been determined, it has been theorized that it is due to binding of SNPs to viral envelope glycoproteins, thereby preventing the viral penetration into the host cell, damage proteins of the viral coat, or fix to the viral DNA [153].

There are not many reports regarding biosynthesized CuNPs that show significant antiviral activity. Below, we describe some reported cases. *Juglans regia* green husk was utilized to biosynthesize CuNPs. The obtained CuNPs in combination with FeNPs resulted in 4.5 TCID50 reduction in viral titer of herpes simplex virus compared with the virus control. It is relevant to mention that the individual antiviral activity of the FeNPs was also evaluated with a reduction of 3.5 times the reduction of the viral titer, demonstrating that the green CuNPs potentiated the antiviral activity. Individually, CuNPs did not show antiviral activity [154]. Another case reported on CuNPs with antiviral activity was that of CuNPs biosynthesized with extracts from *Syzygium alternifolium* fruits, which exhibited antiviral activity against Newcastle disease virus, evaluating and observing positive results in a dose-dependent manner [119].

A particular case of application of CuNPs as an antiviral strategy was the development of textiles whose surface was coated with plant extracts from fruit, leaves, and bark of *Ehretia acuminata* and biosynthesized CuNPs and AgNPs, which can potentially kill Vero-E6 virus when they come in contact with textile surface [115].

Studies conducted on AuNPs and AgNps demonstrate potential for use as antivirals [155]; however, there are few studies regarding CuNps. Until now, the ability of these to bind to the virion, blocking cell receptors, has been proposed by Hang et al., 2015 [156]. Otherwise, it has been proposed that the mechanisms are similar to what is observed in bacteria causing damage by toxic species generated [157,158,159,160]. An interesting application was proposed by Escoffery et al. (2020) impregnating face masks with CuNPs to prevent SARS-CoV-2 infections [161].

Hang et al., in 2015 reported that Cu_2_ONPs can inhibit early-stage hepatitis C infection but lose their activity after 2 h [156]. The three main antiviral mechanisms observed so far are disruption of virus binding to host cells by blocking surface proteins, denaturation of proteins, nucleic acids, and lipids due to the generation of reactive oxygen species (ROS), and disruption of disulfide bonds of viral proteins [106]. However, it is important to continue with these studies to fully determine the effects of CuNPs on viruses.

### 10.5. Antioxidant Activity

Antioxidant activity is associated with various bioreductive phytochemicals on the surface of the CuNPs, as well as their crystal structure, surface charge, particle size, and surface-to-volume ratio. The antioxidant property of CuNPs is attributed to various mechanisms, including inhibition of chain reactions, decomposition of peroxides, binding of transition metal ion catalysts, radical scavenging activity, and inhibition of continued hydrogen abstraction. Free radicals present in the body are unstable and can cause cellular damage by generating reactive oxygen species (ROS) that interact with other molecules during biochemical reactions. CuNPs can absorb, neutralize, and quench free radicals, which contributes to their potential health benefits and protective effects against oxidative stress. Mainly three types of methodologies have been used for the analysis of the antioxidant capacity of NPs. The first estimates the total antioxidant capability (TAC) of the samples by the phosphomolybdenum method, which is based on the reduction of molybdate ions MoO_4_^2−^ (Mo^6+^) into green MoO^2+^ (Mo^5+^) in the presence of NPs in an acid milieu [162]. The second consists of ferric-reducing antioxidant power, which is based on the reduction of Fe^3+^ ions into Fe^2+^ ions through the antioxidant NPs [163]. Finally, the free radical-scavenging activity (DPPH) assay is based on measuring the capacity of antioxidants NPs to scavenge the DPPH radical [140]. Antioxidant activity has been described in CuNPs synthesized with extracts from seeds of *Persea americana*, *Cissus arnotiana*, *Suaeda maritima* (L.) *Dumort*, *Withania somnifera*, and *Phoenix dactylifera* L., among others [164,165,166,167,168]. NPs that have a high antioxidant capacity and at the same time antimicrobial capacity can be considered as potential protectors of host organisms against infections. Antioxidants have been employed to alleviate the adverse consequences of inflammation induced by reactive oxygen species (ROS), particularly in reducing mutations and damage to the host.

### 10.6. Food Packaging

The use of copper-based NPs in the production of packaging materials can extend the shelf life of perishable food products by inhibiting the growth of foodborne pathogens and other spoilage-causing microorganisms. This can potentially improve food safety by reducing the risk of food contamination [169,170].

CuONPs are the most widely used metal oxides in producing packaging materials, mainly because of their antimicrobial properties and potential to inhibit bacterial, fungal, and viral growth [169]. The integration of CuO and CuSNPs into biopolymers made from polysaccharides, proteins, and lipid components is gaining attention as potential biodegradable replacements for conventional plastic packaging films [170]. Remarkably, the incorporation of CuONPs into nanocomposite films of different polysaccharide polymers (agar, alginate, chitosan, among others) not only enhanced the material properties of the polymers but also exerted a strong antibacterial activity against the foodborne pathogens *Escherichia coli* and *Listeria monocytogenes* [171]. Other reports using a polymeric film made of a combination of sodium alginate and cellulose nanowhiskers embedded with CuONPs showed significant antibacterial activity against *Staphylococcus aureus*, *E. coli*, *Salmonella* sp., *Candida albicans*, and *Trichoderma* spp. Moreover, the addition of CuONPs prevented microbial contamination in freshly cut pepper [172].

Although there are many reports on the success of copper-based NPs as potential food packaging materials, there are still concerns about their potential toxicity, especially when used at high concentrations [170]. There is evidence that CuNPs can migrate from packaging matrices to the food contact surface and into the food during the storage and distribution processes. However, the migration properties vary depending on the size, shape, and concentration of the NPs and the type of biopolymers [170,173]. Hence, it is essential to thoroughly investigate the safety aspects of CuNPs in plastic food packaging films, as potential risks to human health are not fully understood.

### 10.7. Wastewater Treatment

CuNPs have important catalytic activity; this characteristic allows them to improve pollutant degradation processes by improving removal or degradation. In wastewater treatment, dyes have a widespread presence because they are generated by different industries. Some synthetic azo dyes like Congo red (CR) have been reported in water bodies. This dye shows several negative effects on human and ecosystem health. For this reason, it is very important to remove these types of carcinogenic contaminants from water. Several methods like photocatalytic degradation have been studied. Chandraker et al., in 2020 reported the complete degradation of congo red in 2 h using CuNPs synthesized with aqueous leaf extracts of *Ageratum houstonianum*, and this photocatalytic behavior was performed at RT and under sunlight [94]. Similar studies found degradation of 70–75% CR with mediated process CuNPs obtained with *Aloe barabdensis* [174]. In a degradation test of malachite green, good catalytic activity of CuNPs mediated by *Galeopsis herba* was reported in water samples [90].

Due to the variety of contaminants present in the water, a wide variety of methods are applied for its treatment. Among them, photocatalysis has positioned itself as a treatment where clean and renewable energy from the sun is used in combination with catalysts such as CuNPs to remove or degrade dyes such as methylene blue [175], Congo red [94], and malachite green [90], among others.

### 10.8. Vegetal Tissue Culture

The micronutrient copper (Cu) plays a vital role in plant growth and tissue culture protocols. Incorporating copper in the form of CuNPs into tissue culture media has demonstrated better outcomes for somatic embryogenesis and regeneration compared to the use of CuSO_4_·5H_2_O [176]. CuNPs are a potential candidate to augment somatic embryogenesis and regeneration of mature embryo explants of wheat. Wheat improvement through callus-based genetic transformation represents a great challenge since this plant is very recalcitrant to the establishment of long-term cultures and finally regeneration of transformed cell lines [177]. A similar effect is observed in other recalcitrant cultivars, like Oryza sativa [178]. Some research studies have reported the beneficial impacts of NPs on callus induction, shoot regeneration, and growth. CuSO_4_NPs significantly increased the shoot length, root length, and fresh weight over control of *Verbena bipinnatifida* seedlings [179]. Furthermore, the treatment of *Solanum lycopersicum* plantlets with green-biosynthetic CuNPs resulted in a substantial increase in their growth and in the increase of sugars and pigments, favoring the organoleptic properties of the fruit [105].

Another potential application for CuNPs is plant micropropagation, to avoid contamination of plants in-vitro and reduced application of antibiotics. NPs are antimicrobial alternatives against a variety of pathogens such as the phytopathogenic fungi *A. alternata* and *F. avenaceum*. Besides, they have positive effects on the improvement of seed germination and plant growth [180].

The demand for naturally occurring bioactive compounds is increasing in the commercial market. Some strategies based on increasing the biosynthesis of phytochemicals during in vitro culture are being developed with CuNPs as a new generation of elicitors. Cell suspension cultures (CSC) of *Gymnema sylvestre* treated with CuONPs resulted in the greatest yields of the metabolites of interest (gymnemic acid II and phenolic compounds), as well as an increase in total phenolics and flavonoids. Additionally, higher antioxidant, antidiabetic, anti-inflammatory, antibacterial, antifungal, and anticancer activities were observed [181].

## 11. Conclusions

With the vast number of medicinal plants that have been used by humanity over time, there is the opportunity to obtain new NPs with different shapes, sizes, and properties to be studied. The possibility of a green, one-step, eco-friendly procedure for the synthesis of Cu-based NPs is attributed to the high and diverse content of metabolites and biomolecules present in plant extracts. The options for obtaining a diversity of phytocompounds are further expanded if we consider that better extraction methods have emerged that allow the stability of the compounds to be preserved. The coating of biomolecules on the surface of NPs renders them more biocompatible when compared to NPs prepared using chemical methods. As the synthesis takes place in the presence of water or ethanol as solvent, the need for toxic solvents harmful to the environment is eliminated.

Regarding the procedure, there are many variables that can be manipulated and that will affect the characteristics and properties of the nanoparticles obtained. A greater comprehension of biochemical components and the molecular mechanisms involved in its synthesis is crucial. To achieve that, it is imperative to identify and isolate compounds that participate in the reduction of metal salts into nanoparticles. On the other hand, detailed investigations are required to be able to scale to an industrial level. Further exploration of the chemical structure of capping agents and active groups present on surfaces is essential to understand the properties of NPs, as well as to identify potential new applications.

There is substantial evidence that highlights the important functionality of copper NPs in various fields of study. Copper is renowned for its eco-friendliness and reasonable cost. The synthesis of Cu-based NPs holds undeniable allure due to their notable antibacterial, antiviral, antifungal, and anticancer properties. Despite the numerous benefits expounded upon in this review, it is essential to acknowledge that exceeding a certain concentration of CuNPs has demonstrated harmful effects on plants, ecosystems, and human beings. Furthermore, conducting comprehensive *in vivo* analyses becomes imperative to assess the treatment’s effectiveness and safety. In addition, this toxicity and risk assessment should be extended to the effective management of the risks associated with their production, use, and disposal. By overcoming these limitations, this technology will bring significant advantages to upcoming generations across various fields.

## Figures and Tables

**Figure 1 micromachines-14-01882-f001:**
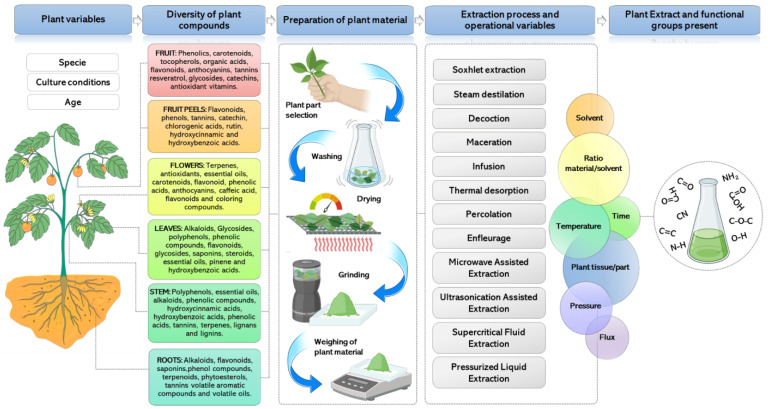
Preparation of plant extracts for nanoparticle synthesis. The diversity of phytocomponents within a given plant can vary both in terms of content and proportion across specific tissues. The general method for the preparation of plant tissues for extraction involves four main steps: material selection, washing and disinfection, drying, and grinding. Subsequently, this material undergoes the extraction process, traditionally performed through decoction. Nevertheless, a wide variety of alternative methods with superior yields and extraction efficiency are available. In all cases, extraction yields can be optimized by manipulating operational variables such as temperature and the duration of solvent extraction, among others. Finally, the plant extract will contain functional groups responsible for the antioxidant, stabilizing and capping capacities of the extracts in the synthesis of NPs.

**Figure 2 micromachines-14-01882-f002:**
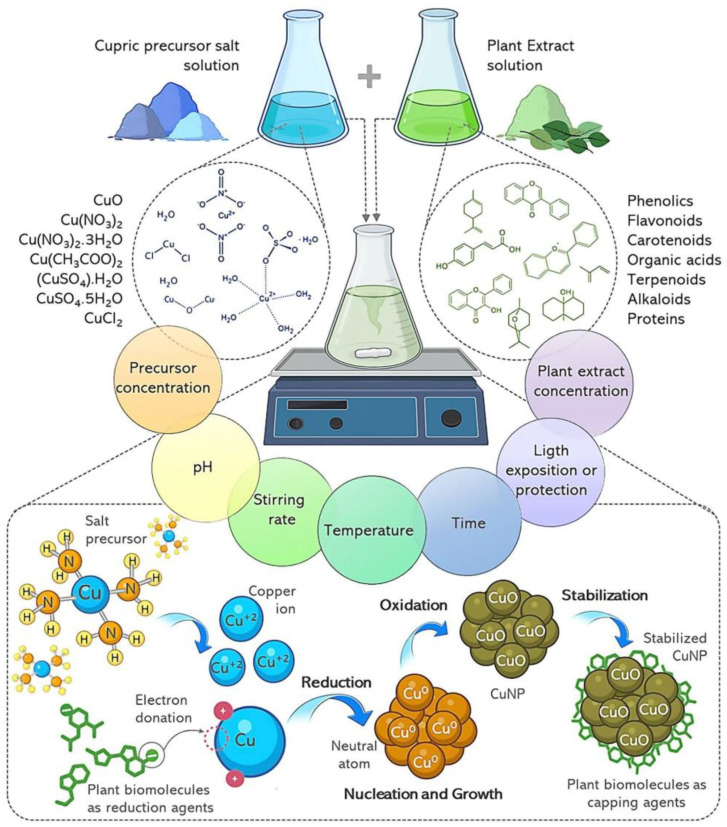
Nanoparticle biosynthesis general process. Schematic diagram showing the general method of CuNP synthesis. Solutions containing metallic copper precursor salts are usually mixed with plant extracts. The compounds that can be present in the synthesis reaction are listed next to the corresponding solution. This process is influenced by various conditions, such as pH, temperature, and precursor-to-extract ratio. The established stages in the green synthesis of NPs are as follows. First, the generation of the ionic form of copper occurs, which can receive electrons from plant compounds with reducing capacity. After the reduction of the ionic form to neutral atoms, small nuclei of copper atoms will aggregate and grow, ultimately resulting in the formation of NPs. Subsequently, these NPs oxidize and are coated through interactions with plant compounds that retain their reactivity. This interaction significantly contributes to the stability of the NPs.

**Figure 3 micromachines-14-01882-f003:**
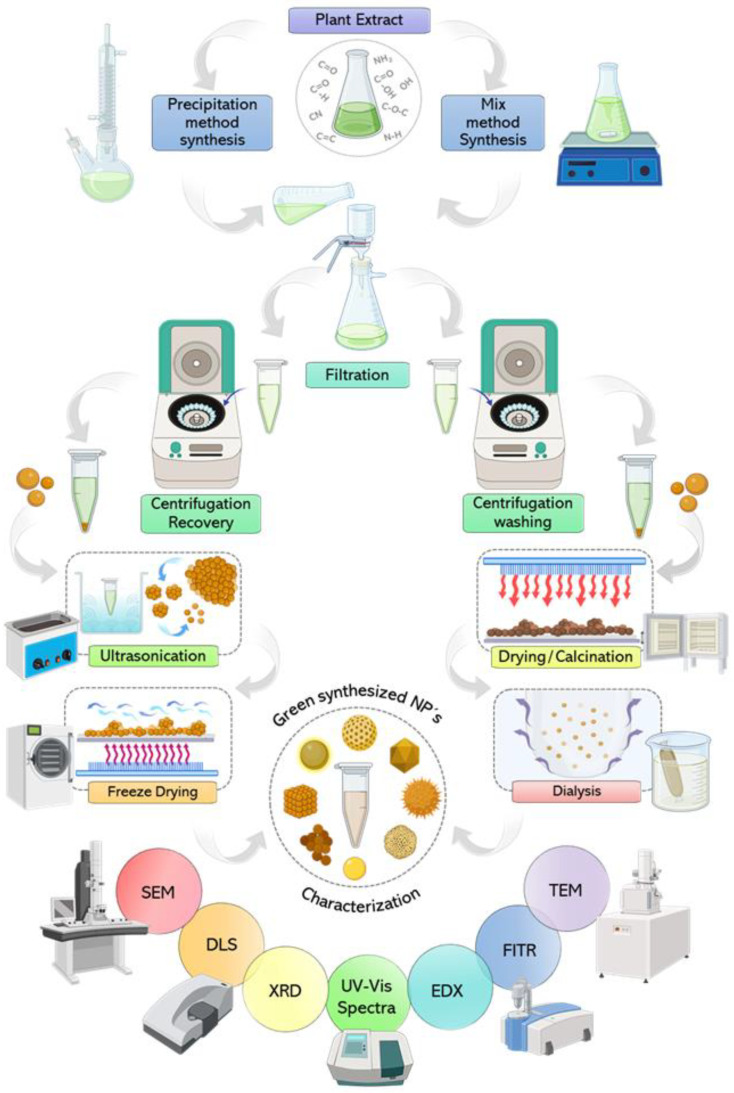
Experimental strategy for NP synthesis and characterization. From the filtered extract, the NP synthesis can be carried out by the mixing method, which is generally carried out at room temperature or by precipitation with the help of an oil bath to maintain high temperatures for a long periods of time. Subsequently, the filtration, recovery, and concentration of NPs can be carried out by means of centrifugation together with various methodologies, such as ultrasonication, lyophilization, calcination, and/or dialysis. NPs can acquire a great diversity of morphologies, which depends on the properties of the plant extracts, among other variables. Analysis by some techniques allows us to know characteristics such as size, and morphology, among other parameters.

**Figure 4 micromachines-14-01882-f004:**
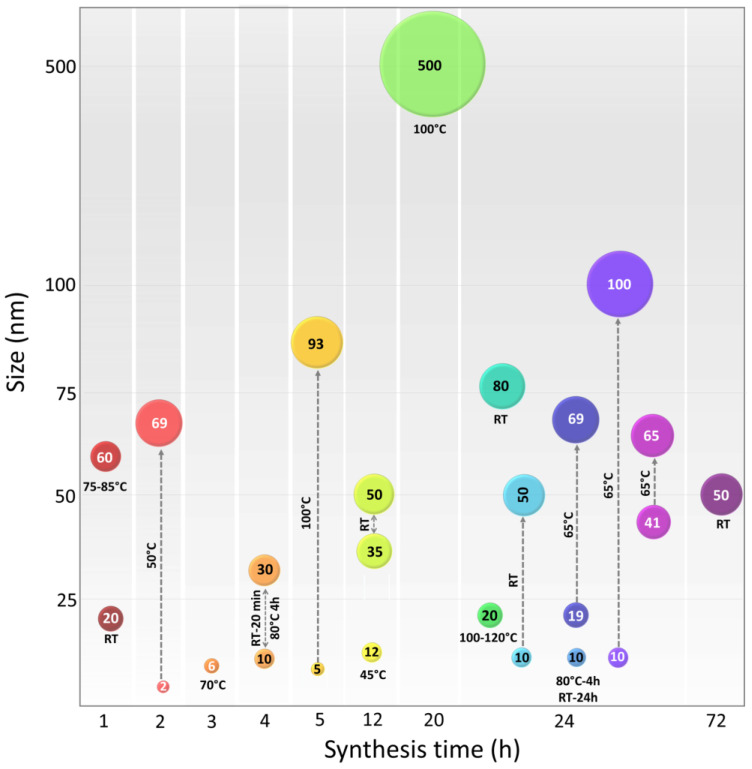
Effect of time and temperature on the size of synthesized NPs. Data from NP synthesis in Table 1 such as temperature and time of synthesis along with the obtained size of NPs are represented in this graph. The size of each sphere is according to the reported size and there is an assigned color for each experiment reviewed. Spheres with the same color that are connected by a dotted arrow indicate the range of NPs obtained in the experiment to which they correspond under those conditions. Additionally, the temperature used in each synthesis protocol can be seen in legends near the spheres or in the arrows. Within each sphere, the size in nm of the size of the NPs is indicated.

## Data Availability

Not applicable.

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
