# Peer review of "Biosynthesis of Copper Nanoparticles with Medicinal Plants Extracts: From Extraction Methods to Applications"

_micromachines, 2023, doi:10.3390/mi14101882_

Round 1

Reviewer 1 Report

The first paragraph is too general and should be more specific to the work being pursued in the manuscript. 

The authors should clarify in their introduction that all of the hazards present in the prior art for the synthesis of nanoparticles are not all present in each synthesis. It should be clarified that these hazards may each be present. 

In the synthesis of nanoparticles within natural or living systems / organisms, the authors should clarify that these presented processes are possible mechanisms, but that there could be other factors at play. Some of their statements are presented as conclusions to how the nanoparticles are formed.

In their historical review, the authors should also clarify that TCM is one of the earliest documented cases of therapeutic remedies. And similarly, the authors should clarify their language to not enter into "history" biases through the use of certain phrases and wording.

As a note, the manuscript should be written in the scientific language and should stick to summarizing the facts and not presenting what could be seen by some as debatable historical views.  The historical review of medicinal plants and the medicinal effects of plant extracts is over emphasized relative to the focus of the manuscript and the length of the manuscript. These parts current detract the attention of the reader.

when reviewing the prior art for the synthesis of cu nanoparticles, summarizing the outcomes in a table might be a better way to represent this data than in paragraph format.

Figure 1 is a great summary. very detailed. The contrast should be improved for ease of reading though. And the authors should check the spelling -- there are some errors therein. 

Figure 2 also has a relatively low resolution.

The summary of techniques for the synthesis of Cu nanoparticles via "biosynthesis" is good. I think the authors should, however, separate out the synthesis of nanoparticles that are done within a plant versus those performed using plant extracts. 

Table 1 is a great summary. there are many errors therein though. Some symbols are incomplete and some values are reported with too high a precision, and some terms are not accurate (filtrated?), and some subscripts are missing, and some chemical formula are split between two lines in an awkward manner, etc. 

The authors should consider grouping the components of Table 1 according to the common salts used (e.g., nitrates, sulfates) to assist in drawing conclusions therein between these commonalities. 

 The authors present an extensive summary of the literature, but a review should also provide insights and forward thinking thoughts for the field. It should include a reflection by the authors of the challenges and opportunities present, beyond just what can be seen from the prior art but to include also the trends and conclusions from the trends observed in their summary. 

The authors should consider further figures to assist the reader, including figures showing representative data that is relevant to the main focus of the review. 

Spelling errors noted in part, but are relatively minor. And some formatting errors in the references. 

Reviewer 2 Report

This contribution aims to review different methodologies to biosynthesise copper nanoparticles with medicinal plant extracts and survey their applications.

The general level of English in the contribution is well written, some minor phrasal problems were found and a native-English speaker proofreading is highly suggested in order to avoid these details. In particular, a few grammar details between the use of singular and plural required to be revised, just some examples of the latter were given, but many others are present:

On page 2, lines 58-60 “In this review, we present an overview of the green synthesis method applied to Cu-NPs, focusing on reported plant extracts used for NP-synthesis, mainly from medicinal plants.” E.g there is a redundancy for “synthesis”, and also, perhaps should be “green synthesis methods” or just one method has been applied??

Another example is present in line 264 “could be effectively mitigate and there are an enhance of the knowledge of our scientific”, please rephrase accordingly.

Also in lines 290-291 “function of chemical defence against”, please correct.

In line 311 “new biomolecules to be used in the synthesize of NPs.”, please correct.

Avoid the use of “element” for plant means, e.g. in lines 313-314 “since the plant element has different richness and distribution 313 of the metabolites of interest”.

In line 83 “in vivo” please use italics.

In lines 129-131 you need to revise these phrases since due to separation they seem disconnected and the meaning is fuzzy, please reconnect those phrases in order to recover the original meaning “In each solution, the type of compounds that may be present in the synthesis reaction are shown in a representative way. By action of the conditions to which the synthesis is submitted, such as pH, temperature, precursor: extract ratio, etc.”.

According to what was written in lines 278-282 “despite the knowing of the redox mechanisms of these functional groups, there is still no fully elucidated mechanism for the synthesis of NPs. Nevertheless, due to the great variety and richness of biomolecules that there may be part of plant extracts, there is still no clearly elucidated which plant-derived compound is specifically responsible for the reducing, oxidative, capping, or stabilizing properties.”, it has been stated that a very small review related to particular molecular entities that are able to develop metallic NPs must be included, since e.g. ascorbic acid, chitosan, among others, are well-known systems to develop such reduction capacities required for this means. Please follow the next DOIs in order to make notice of these important results for your particular needs, just as examples of the requested task.

https://doi.org/10.1039/C6RA00194G

https://doi.org/10.1016/j.jksus.2022.101927

https://doi.org/10.1016/S1003-6326(11)61449-0

https://doi.org/10.1016/j.carbpol.2014.05.081

https://doi.org/10.3390/molecules26102968

https://doi.org/10.1016/j.ijbiomac.2019.11.179

https://doi.org/10.1016/j.arabjc.2021.103259

And also as you have stated in lines 298-299 “Hydroxamic acid and compounds having a benzamide group have potential as anti-cancer drugs [66].”, these hydroxamic acids are well-known siderophores, but they also are capable of NP development in general by redox procedures.

In figure 2 you state that the plant biomolecules serve as NP-capping agents or stabilizers, is this the only function that these molecules provide to the system?? or there are other important characteristics resulting in the NPs with the incorporation of these capping moieties?? Please revise.

Additionally, it has been considered that a more thorough structure-property-activity-like correlation and analysis section is missing, since depending on the type of extract, the type of Cu source (salt), and reaction conditions, would develop different sizing, as well as Cu/O ratios, among other characteristics in the resulting NPs. Please try to develop this missing part, you have enough material in order to provide these important correlations at least in a qualitative way.

Also are missing the best two or three procedures that were more important according to the characteristics that you have been underlining during the whole contribution, these are i) the stability of the resulting NPs, ii) the greener method, and iii) perhaps the best biological activity, but if you consider other, please include this/them. In this sense please provide at least one procedure that aims for each one or preferably two of these characteristics, and conclude about these findings as well. This part could also be included in the requested new section of “structure-property-activity-like correlation“, where you should notice this while the development of the section is being carried on.

The general level of English in the contribution is well written, some minor phrasal problems were found and a native-English speaker proofreading is highly suggested in order to avoid these details. In particular, a few grammar details between the use of singular and plural required revision, some examples of the latter were given to the authors but general proofreading should be done to enhance quality.

Reviewer 3 Report

The review manuscript entitled, “Biosynthesis of copper nanoparticles with medicinal plants extract: from extraction methods to applications” submitted by Aurora Antonio-Perez et al. to the journal Micromachines. The way authors discussed the review was more interesting and has covered almost all over the proposed area of interest. My recommendation toward this review manuscript to the Micromachines journal is “Minor Revisions”.

  1. Lack of discussion/figures on mechanistic aspects of copper nanoparticles against bacteria and viruses.
  2. Few areas observed incomplete sentences. Need to correct English language carefully.

 Authors discussed the proposed topics in a systematic manner and properly cited the relevant references wherever required. Overall the manuscript is suitable for publication after addressing the above comments.

English language used in this review manuscript was clearly readable and easy to understand.

Round 2

Reviewer 1 Report

The authors have addressed my concerns. The revised manuscript has been improved and clarified, and is ready to proceed with publication. A careful read through is warranted for clarity and conciseness. 

As mentioned above, focus your final edits on clarify and being concise.